# Microwave Absorption Properties of Multi-Walled Carbon Nanotubes/Carbonyl Iron Particles/Polyurethane Foams

**DOI:** 10.3390/ma15165690

**Published:** 2022-08-18

**Authors:** Xuegong Huang, Danping Yu, Simin Wang

**Affiliations:** School of Mechanical Engineering, Nanjing University of Science & Technology, Nanjing 210094, China

**Keywords:** microwave absorbing material, microwave absorption property, CST electromagnetic amplitude simulation, illumination experiment, carbon nanotube

## Abstract

In order to improve the microwave absorption performance of absorbing materials, the composite foam absorbing materials with different multi-walled carbon nanotube (MWCNT) contents were prepared using polyurethane foam as the substrate and MWCNTs and flaked carbonyl iron powder as absorbers. The electromagnetic properties of the materials were characterized and analyzed. Then, CST electromagnetic simulation software was used to simulate the electromagnetic shielding effect of absorbing materials on mechatronics products under a strong electromagnetic irradiation environment, and, finally, it was verified by irradiation experiment. The results show that the materials have good microwave absorption properties, in which the composites containing 1.5 wt.% MWCNTs exhibit good microwave absorption properties. The minimum reflectivity reaches −29 dB when the thickness is 3 mm and −15.6 dB when the thickness is 1.5 mm, with a bandwidth of 5.7 GHz for reflectivity less than −10 dB. The good microwave absorption performance of the material is due to the synergistic effect of MWCNTs particles and good impedance matching. The simulation and experimental results show that the mechatronics product with absorbing materials can protect against strong electromagnetic interference and ensure the normal operation of the mechatronics product circuits.

## 1. Introduction

Microwave absorbing materials (MAMs) have broad application prospects in the fields of electromagnetic pollution prevention and control, signal interference shielding, radar detection and counter-detection, aerospace, military equipment, etc. At present, a relatively mature theory and application system has been formed for absorbing materials. Various absorbers with complex permittivity and complex permeability have been developed to design MAMs. According to the composition, it can be classified as carbon-based absorbing material, iron-based absorbing material, ceramic-based absorbing material, and other types of absorbing material [1,2,3]. However, with the growing demand and increasing requirements, the problems of a narrow absorbing band and high density of existing absorbing materials are gradually exposed. In this context, carbon-based composites have been extensively studied as lightweight flexible coating materials for electromagnetic shielding [4,5,6]. Among them, new materials, such as carbon nanotubes and graphene, have attracted much attention. Graphene is a two-dimensional layered carbon nanomaterial with a high dielectric constant, high thermal conductivity, high electron mobility, and large specific surface area. When graphene is used alone as an absorbing material, the absorption mechanism is relatively simple and the absorbing performance is limited. Therefore, compounding with other materials is the main trend in the development of graphene-based absorbing materials [7]. Jasvir Dalal et al. synthesized PEDOT/reduced graphene oxide (RGO)/PbTiO_3_ nanocomposites by a facile in situ chemical oxidative polymerization method. Experimental measurements show that an enhanced electromagnetic shielding effectiveness value of 51.94 dB (>99.999% attenuation) has been achieved in a 12.4–18 GHz frequency range [8]. In addition, Jasvir Dalal et al. used a facile in situ synthesis method to prepare Ni_0.5_Co_0.5_Fe_2_O_4_/RGO nanocomposites. The results show that, in the frequency range of 12.4–18 GHz, when the corresponding thickness is 2 mm, the shielding effectiveness reaches 36.3 dB (electromagnetic wave attenuation is about 99.98%) [9]. Luo et al. proposed a facile one-step solvothermal reduction method for the fabrication of RGO/FeNi_3_/Fe_3_O_4_ MAMs. When the material thickness was 1.9 mm, the minimum RL was up to −46.6 dB at 12.5 GHz and the absorption bandwidth was 9.26 GHz [10].

Multi-walled carbon nanotubes (MWCNTs) are of great interest for their light weight, high electrical conductivity, large specific surface area, and strong dielectric loss [11]. Single MWCNTs are difficult to use in the field of microwave absorption, and their superconductivity can cause skin effects, making it difficult for electromagnetic waves to enter the material. Further, the lack of magnetic loss characteristics leads to poor impedance matching, which, in turn, affects its wave absorption capability. While further exploring the absorption mechanism of MWCNTs, researchers often compound MWCNTs with semiconductor dielectrics, magnetic materials, and polymers to provide the composite materials a variety of loss mechanisms [12,13]. For example, Xing et al. prepared CoFe_2_O_4_/MWCNTs composites by co-precipitation method and tested the electromagnetic parameters of the material with a network vector analyzer. The results show that, when the thickness of the material is 2 mm, the maximum absorption value of CoFe_2_O_4_/MWCNTs is −32.5 dB at 8.88 GHz, and the bandwidth of less than −10 dB in the whole frequency band is 2.4 GHz. [14]. Bi et al. prepared MWCNTs/polyaniline/cobalt composite microwave absorbing materials by in situ polymerization and electroplating process, and the excellent microwave absorbing properties were obtained at a filling ratio of only 5% in paraffin due to the good electrical conductivity of both MWCNTs and polyaniline and the addition of magnetic metal cobalt [15]. Tang et al. successfully synthesized MWCNTs/zinc ferrite composites by co-hydrothermal and sol-gel methods, and the multi-material components provided the composites interfacial and dipole polarization, eddy current loss, conductive loss, etc. The material has a good low frequency microwave absorption performance of −0.65 dB at 0.81 GHz, and the effective absorption bandwidth is 0.45–1.42 GHz at only 2.5 mm [16]. Mariya A. Kazakova et al. successfully prepared a set of Co/MWCNT-supported polyethylene composites with different compositions by in situ polymerization of ethylene. The electromagnetic properties of the composites were tuned by varying the filler loading and Co:MWCNT ratio. The results show that the Co/MWCNT-PE composite with a total filler and Co loading of only 12 and 1.7 wt.%, respectively, showed extremely high reflection loss (RL) of −55 dB. More importantly, an effective bandwidth of 12.8–17.8 GHz (RL below −10 dB) was achieved for a matching thickness of only 1.5 mm [17]. Gagan Deep Aul et al. used a three-step method to prepare materials with different weight percentages of cobalt sulfate and MWCNTs. The study shows that the maximum reflection loss observed (50% MWCNT and 50% CobaltSulphate) is −30.22dB at 11.8GHz and the maximum bandwidth window available (40% MWCNT and 60% CobaltSulphate) is 3.9 GHz in the frequency range of 8–13 GHz with 3 mm thickness [18]. There is still much literature on the research of carbon matrix composites. Table 1 summarizes and analyzes the research status in recent years. It can be seen from the table that MWCNTs are ideal absorbents for the preparation of new composite absorbing materials. Therefore, in this paper, MWCNTs are used as absorbing materials; other materials are composited to prepare new absorbing materials that are lighter and thinner and have better absorbing effects.

With the advantages of high saturation magnetization strength, high permeability, high Curie temperature, and good mechanical properties [21], flake carbonyl iron (FCI) particles have become one of the earliest researched and most widely used wave absorbing materials at home and abroad. However, in actual use, FCI has difficulty meeting the demands of light weight and wide absorption of microwave absorbing materials due to its high density and single loss mechanism [22]. Therefore, it also needs to be combined with other absorbers to improve the overall absorption performance of the material. Xu et al. separately dispersed SCI/FCI with CNTs in silicone rubber using a double-roller machine and showed that CNTs compounded with FCI could achieve a larger absorption broadband and higher absorbance, with a minimum reflectance of −35 dB at 3.5 GHz when the thickness was 2 mm. The effect was attributed to the directional distribution of FCI caused by the interaction between CNTs and FCI [23]. Qing et al. investigated the microwave absorption performance of FCI as a magnetic absorber and carbon fiber (CF) as a conductive absorber filled epoxy silicone resin coating. The results show that, when the content of FCI is 65 wt.%, CF is 2 wt.%, and the coating thickness is 1 mm, the reflectivity is below −10 dB in the frequency range of 8 to 18 GHz [24]. From the current state of research, FCI absorbers will remain the main direction of research and application of microwave absorbing materials, and the effective compounding of FCI with other types of absorbers is an effective means of improving wave absorption performance. In this paper, MWCNTs are compounded with FCI to create absorbing material with both electric and magnetic losses while effectively broadening the absorbing frequency band and improving the absorbing performance. In addition, polyurethane (PU) foam is light, high-strength, has oxidation resistance, low temperature stability, wear resistance, and strong adhesion to inorganic materials [25], etc. After the absorber is compounded with it, the porous nature of the PU-based foam allows the polymer-based composite to have a characteristic impedance close to that of air, which helps to reduce the reflection of electromagnetic waves on the surface of the material after the absorber is compounded with it. On the other hand, the good impact resistance of the PU foam makes the composite material resistant to high overload.

This paper presents a flexible porous composite absorbing material with MWCNTs and FCI particles as absorbers and PU foam as the matrix. The complex permittivity and complex permeability of the composites were measured by the T/R method and the electromagnetic and microwave absorption properties of the materials were investigated in detail. In addition, to better illustrate the effect of preparing MWCNTs/FCI/PU composite absorbing materials in a practical application environment, the three-dimensional electromagnetic simulation software CST was used to simulate the absorbing performance of the composite materials under strong electromagnetic wave interference, finally verified by amplitude illumination experiments. This paper analyzes all the aspects of composite materials from material preparation, testing, simulation, and experimentation, which is the highlight of this article.

## 2. Sample Preparation

### 2.1. Material Preparation

The chemicals and materials used for the preparation of MWCNTs/FCI/PU composite absorbing materials are shown in Table 2. The main equipment includes ultrasonic disperser, vacuum drying oven, precision booster electric stirrer, and electronic balance FA1004N.

PU is produced by the addition polymerization reaction of polyether polyol and polyisocyanate. Resin dispersant and ultrasonic disperser are used for the dispersion of MWCNTs (outer diameter: 8~15 nm, length: 3~12 μm, purity: >95%) in polyether polyol to prevent MWCNTs from agglomerating in the system. The stirrer is mainly used to promote the dispersion of FCI (α-iron content >99.5%, diameter 1–5 μm, thickness < 1 μm) in the system and full reflection of polyether polyol and polyisocyanate. The drying oven provides a constant temperature environment for the maturation of the foam material. The electronic balance has a division value of 0.01 g and is used for weighing each component. Preparation of samples by adding different components of MWCNTs and FCI powder to PU. The detailed composition of the samples is shown in Table 3.

Adequate homogeneous dispersion of MWCNTs and FCI is required for adequate absorption performance of the absorber. Firstly, polyol and resin dispersant were mixed thoroughly, then MWCNTs were added for full dispersion at 25 °C environment using ultrasonic disperser to obtain the dispersion slurry. Then, add the FCI into the dispersion slurry and adjust the electric stirrer speed to 2000 rpm to stir the mixture for 30 min. The polyisocyanate is then added to it and continued to be stirred until the mixture is homogeneous and injected into the ring mold to allow free foaming. Finally, the foamed material is put into the vacuum drying oven (35 °C) to mature for 2 h. After maturation, the specimens were polished and machined into a circular shape with an outer diameter of 7.0 mm, an inner diameter of 3.04 mm, and a length of approximately 3 mm, and this is to match the test system.

### 2.2. Material Characterization and Measurement

To test the wave absorption effect of the composite material, its complex permittivity ($\varepsilon =\varepsilon^{\prime }-j\varepsilon^{\prime \prime }$) and complex permeability ($\mu =\mu^{\prime }-j\mu^{\prime \prime }$) were measured by T/R coaxial line method with a network analyzer (Agilent technologies E5071C:100 KHz~20 GHz), the frequency range of 2 to 18 GHz. The basic principle of the measurement is that the absorbing material sample placed in the coaxial transmission line is equivalent to a two-port network, and the amplitude and phase shift of the reflected and transmitted signals when the electromagnetic wave is incident on the sample are measured, thereby calculating the electromagnetic parameters of the material.

The reflectance (RL) of the absorbing material can be calculated using Equations (1) and (2) [26].
(1)$$\mathrm{RL}\;(\mathrm{dB})=20\log\left|\frac{\left(Z_{\mathrm{in}}-Z_{0}\right)}{\left(Z_{\mathrm{in}}+Z_{0}\right)}\right|$$
(2)$$Z_{in}=Z_{0}\sqrt{\frac{\mu }{\varepsilon }}\tanh\left\{j\left(\frac{2\pi \mathrm{fd}}{c}\right)\sqrt{\mu \varepsilon }\right\}$$
where $Z_{in}$ denotes the impedance of the material to a vertically incident plane wave, $Z_{0}$ denotes the impedance of free space, c denotes the speed of light, d denotes the material thickness, and f denotes the frequency of the incident electromagnetic wave.

## 3. Electromagnetic Performance

### 3.1. Electromagnetic Parameters

The microwave properties of MWCNTs/carbonyl iron powder/polyurethane foam composite absorbing materials are evaluated from the two electromagnetic property parameters of complex permittivity ($\varepsilon =\varepsilon^{\prime }-j\varepsilon^{\prime \prime }$) and complex magnetic permeability ($\mu =\mu^{\prime }-j\mu^{\prime \prime }$). The results of four groups of samples are shown in Figure 1.

From Figure 1a,b, it can be seen that the value of the real part of the dielectric constant *ε*′ decreases with increasing frequency for the four groups of samples, where the decreasing trend is more obvious for S1.5 and S2 groups. In addition, the dielectric constant real part *ε*′ and imaginary part *ε*″ become larger with the increase in MWCNTs. According to the research, the dielectric constant of microwave materials is mainly related to the properties of the matrix, the composition of the absorber and the relative content of each component, and the structure of the composite material [27,28,29]. The interface of MWCNTs/FCI/PU composite absorber increases with the increase in MWCNTs content. Under the action of external high-frequency electromagnetic waves, the migration and accumulation of charges on the interface also increase, and the interface polarization increases, resulting in an increase in the real part of the dielectric constant. According to the free electron theory [30]: $\varepsilon^{\prime \prime }\approx 1/(2\pi \varepsilon_{0}\rho f)$, ρ is material resistivity. MWCNTs have high specific surface area and electrical conductivity. When the relative content of MWCNTs increases, it is easy to form a large conductive network with polyurethane matrix, thus reducing the resistivity of the composite. Therefore, the dielectric constant increases with the increase in the relative content of carbon nanotubes. These results are also consistent with some previous studies on carbon-based composite absorbing materials [31].

Figure 1c,d shows the influence of the change of the content of MWCNTs on the material permeability when the carbonyl iron powder content remains unchanged. It can be seen from the figure that both the real and imaginary parts of the magnetic permeability show different degrees of increase with the content of MWCNTs. When the mass fraction of MWCNTs is 1.5 wt.%, the real and imaginary parts of the magnetic permeability of the composites exhibit the maximum value. For the composites synthesized with MWCNTs and FCI as absorbents, the complex magnetic permeability is directly influenced by the adsorbent (MWCNTs and FCI) content and the interfacial boundary conditions. The MWCNTs/FCI/PU interfacial characteristics not only affect the interaction between MWCNTs, FCI, and the surrounding matrix but also have an effect on the exchange bias and magneto-crystal anisotropy of FCI in some special cases [28]. According to previous studies, the interaction between carbon nanomaterials and magnetic particles can significantly change the magnetic properties of the material [32,33,34]. In addition, under strong electromagnetic wave amplitude, the closed loop formed by the distribution of MWCNTs along the vesicle walls generates induced currents, which causes the generation of magnetic moments in MWCNTs similar to the magnetic moment effect in magnetic materials [35], which also leads to an increase in the material complex permeability.

The dielectric tangential loss shows the energy dissipation due to the external electric field in the form of motion and heat, and the magnetic tangential loss shows the energy loss inside the magnetic material due to the phase delay between the applied and induced magnetic fields [36]. When frequencies vary from 2 to 18 GHz, the dielectric loss and magnetic losses of the material are shown in Figure 2. It can be seen from the figure that the material dielectric loss tangent increases with the content of MWCNTs, and there is also a small tendency to increase with frequency, where samples S1.5 and S2 have better performance, i.e., better dielectric loss capability. Regarding Figure 2a,b, it can be seen that the magnetic loss of the samples is greater than the dielectric loss; therefore, the performance of the composite absorbing material mainly depends on the magnetic loss, and composite sample S1.5 shows the best electromagnetic wave absorption performance.

### 3.2. Microwave Absorption Performance

Figure 3 shows the variation curves of reflectance of MWCNTs/FCI/PU composite absorbing material with the content of MWCNTs. The figure shows, when the thickness of the material is 3 mm, the minimum reflectance of sample S0 is −12.3 dB, which is less effective than other samples in absorbing waves. The overall absorption performance is improved when the MWCNTs content is increased to 1 wt.%, and the reflectivity reached −10 dB in the frequency range of 6.9~10.5 GHz, with a minimum reflectivity of −15.9 dB. S1.5 has the best reflectance among the 4 samples, with a minimum reflectance of −30 dB at 5.2 GHz. When the content of MWCNTs increases to 2 wt.%, the minimum reflectivity is −22.5 dB, which shows that the minimum reflectivity of the composite does not continue to increase with the increase in MWCNTs content. It can also be seen from the figure that the maximum absorption peak frequency decreases with increasing MWCNTs content, from 11.3 GHz at 0 wt.% to 4.8 GHz at 2 wt.%.

Impedance matching and attenuation constant are two key factors for absorbing materials. The material impedance matching ratio |*Z*| and the attenuation constant α can be calculated according to the following Equations [37,38], and the different samples |*Z*| and *α* curves are shown in Figure 4.
(3)$$\left|Z\right|=\left|\frac{Z_{in}}{Z_{0}}\right|$$
(4)$$\alpha =\frac{\sqrt{2}\pi f}{c}\sqrt{\left(\mu^{\prime \prime }\varepsilon^{\prime \prime }-\mu^{\prime }\varepsilon^{\prime }\right)+\sqrt{{\left(\mu^{\prime \prime }\varepsilon^{\prime \prime }-\mu^{\prime }\varepsilon^{\prime }\right)}^{2}+{\left(\mu^{\prime \prime }\varepsilon^{\prime \prime }+\mu^{\prime }\varepsilon^{\prime }\right)}^{2}}}$$

In Equation (3), $Z_{in}$ and $Z_{0}$ are the impedance of the absorbing material and that of free space, respectively. In Equation (4), *f* is frequency and *c* is the velocity of light. The characteristic of impedance matching refers to the ability of the incident wave to enter the absorbing material. Generally, the closer the value of |*Z*| to 1, the better the impedance match of the sample [39], and the electromagnetic waves are more easily incident inside the material and attenuated. The attenuation constant is used to measure the attenuation ability of a material. Figure 4 shows, compared with S0, the impedance matching characteristics of samples S1.5 and S2 are closer to 1, and, at the same time, they have higher attenuation constants, so they have strong wave absorption performance.

Based on the previous electromagnetic performance analysis of four groups of samples, the best performance of S1.5 sample was compared comprehensively. The following analysis was conducted on the variation in reflectivity with thickness for the MWCNTs content of 1.5 wt.% and FCI content of 70 wt.% filled PU absorbers. The material thickness was increased from 1 mm to 3 mm in five experimental groups, and the results are shown in Figure 5. The figure shows that the maximum absorbing frequency of the material decreases from >18 GHz at 1 mm to 5.2 GHz at 3 mm. The minimum reflectivity also decreases with increasing thickness and reaches −29 dB at 3 mm. The reflectivity can be less than −10 dB in the frequency range of 9~14.7 GHz, 7~11.7 GHz, 4.7~7.7 GHz, and 4.1~6.5 GHz when the thickness is 1.5, 2, 2.5 mm, and 3 mm.

## 4. Simulation of Interference Suppression Effect of Microwave Absorbing Materials

In order to verify the electromagnetic protection performance of the absorbing material prepared, a mechatronics product is taken as the research object and the anti-electromagnetic interference performance of the mechatronics product is analyzed with or without absorbing material. The electromagnetic interference pulses include high power microwave (HPM), ultra-wide band pulse (UWB), and high-altitude nuclear electromagnetic pulse (NEMP). The mechatronics product approximation model in CST Microwave Studio is shown in Figure 6. The model has a diameter of 50 mm, a height of 50 mm, a wall thickness of 2 mm, and is made of aluminum. A circular hole of 4 mm in diameter is cut in the top center of the cavity to serve as an over-hole for connecting the detector to the low-frequency circuit. The low-frequency circuit board was replaced by a macroscopic dielectric plate [40] with a conductivity of *σ* = 0.22 S/m and a relative permittivity of *ε* = 2.65, with dimensions of Φ = 32 mm and a thickness of 1 mm, placed 8 mm from the bottom of the housing. The whole model adopts an xyz three-dimensional right-angle coordinate system, and the coordinate origin is located at the center of the bottom surface of the cavity. The electromagnetic interference source is incident as a line-polarized plane wave in the x direction to simulate far-field interference with the incident direction of −z.

In order to study the comprehensive protection effect of microwave absorbing material, the field intensity distribution inside the product before and after the placement of microwave absorbing material under the action of different strong electromagnetic energy interference sources (HPM, NEMP, UWB) is simulated. To calculate the coupling effect of the interfering source on the product cavity before and after placing the absorbing material, the coupling coefficient *η* defined in the literature [41] is introduced.
(5)$$\eta =20lg\left|\frac{E_{c}}{E_{i}}\right|$$

*E_c_* represents the electric field strength at that point inside the cavity, and *E_i_* represents the incident source electric field strength. The larger the coupling coefficient, the more the product is influenced by the interference source and the poorer the material interference suppression effect is.

### 4.1. High-Power Microwave (HPM)

Microwaves with peak pulse power greater than 100 MW are called high-power microwaves. HPM electromagnetic pulse signal can be generally expressed by the following Equation (6) [42].
(6)$$E\left(t\right)=\{\begin{array}{c} E_{0}\frac{t}{t_{r}}\sin\left(2\pi f_{0}t\right),\\ E_{0}\sin\left(2\pi f_{0}t\right),\\ E_{0}\left(\frac{t_{d}+2t_{r}}{t_{r}}-\frac{t}{t_{r}}\right)\sin\left(2\pi f_{0}t\right), \end{array}\begin{array}{c} 0<t\leq t_{r}\\ t_{r}<t\leq t_{r}+t_{d}\\ t_{r}+t_{d}<t\leq 2t_{r}+t_{d} \end{array}$$
where *E*_0_ is the peak pulse signal, *f*_0_ is the center frequency, *t* is the pulse excitation time, *t_r_* is the pulse rise time, and *t_d_* is the pulse width. The field strength *E* of an HPM at a distance of *R* meters from the emitting source in a low-altitude atmospheric environment can be expressed by Equation (7) [43].
(7)$$E=\frac{\sqrt{60P_{t}G_{t}}}{R}$$
where *P_t_* is the transmitting power and *G_t_* is the transmitting antenna gain. Set the distance between the product and the HPM transmitting source as 200 m, the transmitting power as 1 GW, and the gain of the transmitting antenna as 20 dB. *E*_0_ = 12.25 kV/m is calculated from Equation (7), setting *f*_0_ = 5 GHz; *t_r_* = 5 ns; *t_d_* = 10 ns. The time and frequency domain waveforms of the HPM pulse are shown in Figure 7, from which it can be seen that the HPM pulse energy is mainly concentrated at the center frequency of 5 GHz.

Figure 8 shows the change of the coupling coefficient before and after placing the absorbing material obtained from the simulation. From Figure 8, it shows that the coupling coefficient is much greater than zero in some frequency range when the absorbing material is not affixed. This indicates that the electric field is not attenuated but enhanced. This is because the electromagnetic waves entering the cavity cannot be effectively dissipated and oscillate back and forth in the cavity, causing the resonance to produce a field strength enhancement effect. After mounting microwave absorbing material, the coupling coefficient is smaller in all aspects, and the coupling coefficient is less than zero, indicating that the field strength enhancement effect is suppressed. This is consistent with the study in the literature [44].

Figure 9a,b shows the absolute value of electric field intensity at P (0, 0, 25) before and after placing the absorbing material time from 0 to 30 ns. As can be seen from Figure 9a, when there is only a metal shell without absorbing materials, the peak value of the interference field in the cavity decays to 168 V/m, and, when the eternal radiation interference disappears, the response interference electromagnetic field in the metal shell will continue for a period of time. This is because the electromagnetic interference energy cannot be effectively lost inside the cavity. If the product is continuously irradiated by HPM, the electric field inside the system housing will be continuously superimposed and the peak field strength will greatly exceed 168 V/m. Figure 9b shows that, after placing the absorbing material, the peak field strength decays to 2.4 V/m under the joint action of the shell and absorbing material. At this time, the electric field will disappear simultaneously with the radiation interference source, which greatly eliminates the influence of the HPM pulse.

### 4.2. Ultra-Wideband Pulse (UWB)

Ultra-wideband pulse refers to the pulse signal of peak output power above 100 MW, the rising front time of the pulse is at the picosecond level, and the pulse bandwidth is greater than 25% of the electromagnetic radiation. The more commonly used UWB signals are unipolar pulses, linear FM pulses, and Gaussian pulses. In this section, Gaussian pulses are chosen as UWB pulse signals. The Gaussian pulse is shown in Equation (8) [45].
(8)$$E_{t}=E_{0}e^{\frac{-4\pi {\left(t-t_{0}\right)}^{2}}{\tau^{2}}}$$
where *E_t_* is the field strength of the Gaussian pulse, *E*_0_ is the amplitude voltage of the UWB pulse, *t*_0_ is a constant that determines the moment of peak pulse appearance, and *τ* is a constant that determines the pulse width. Figure 10 is a UWB waveform diagram, where *t*_0_ = 0.25 ns and *τ* = 0.0483.

As shown in Figure 11, it can be found that the shell has a certain electromagnetic shielding effect on the UWB pulse regardless of whether there are absorbing materials because the energy of the UWB pulse signal is not concentrated and the energy of the unit frequency band is sparse. The suppression effect of the absorbing material on the interference of the UWB is worse than that on the HPM, but it also effectively suppresses the electromagnetic resonance in the cavity.

Figure 12a,b shows the absolute values of electric field intensity at P (0, 0, 25) before and after applying absorbing material under ultra-broadband pulse irradiation time from 0 to 2 ns. The figure shows that the electromagnetic energy inside the shell is poor. The peak electric field intensity at the P point is 20 V/m, and the electric field does not disappear simultaneously with the radiated interference source but continues to exist for a period of time. After placing the absorbing materials, the field strength peak attenuation will reach 10 V/m and the electric field will drop to below 2 V/m in a very short period of time, effectively suppressing the effect of HPM pulse on the system.

### 4.3. High Altitude Nuclear Electromagnetic Pulse (NEMP)

NEMP is a transient electromagnetic radiation that accompanies a nuclear explosion, which is another nuclear explosion effect of nuclear weapons in addition to shock waves and thermal radiation. This section uses the NEMP waveform defined by the IEC standard, defined as a double exponential pulse, and the function expression is Equation (9) [46].
(9)$$E\left(t\right)={kE}_{0}\left(e^{-\alpha t_{r}}-e^{-\beta t_{f}}\right)$$
where *E*_0_ is the peak field strength, *k* is the correction factor, *α* and *β* are the front and back edge parameters of the pulse, *t_r_* is the rising edge duration, and *t_f_* is the half-width height duration. The IEC standard specifies the NEMP waveform parameters with *α* = 4 × 10^7^, *β* = 6 × 10^8^, *k* = 1.3, *E*_0_ = 50 kV/m, rising edge duration *t_r_* = 2.5 ns, and half-width high duration *t_f_* = 55 ns. The nuclear electromagnetic pulse waveform is shown in Figure 13 [43]. From the spectrum diagram, we can see that the electromagnetic energy of NEMP is mainly concentrated below 1 GHz, where the electromagnetic energy starts to decay from about 10 MHz to almost 0 at 1 GHz.

As shown in Figure 14 and Figure 15, it can be found that HEMP has less effect on the product. In the case of no absorbing material, the peak value of the electric field of the nuclear electromagnetic pulse interference source is 50 kV/m, and the peak value of the electric field coupled into the cavity is only 0.48 V/m. At this time, only relying on the shell can demonstrate a good electromagnetic shielding effect. After placing the absorbing material, the field strength peak is 0.23 V/m, and it decays to 0.1 V/m in a short time; placing absorbing material can make the coupling coefficient reduce by about 20 dB.

## 5. Experimental Study on Anti-Electromagnetic Radiation for Absorbing Material

In Section 4, the comprehensive protection effect of microwave absorbing material on the mechatronic product under the action of different strong electromagnetic energy interference sources is investigated by the electromagnetic simulation software CST. In order to verify the theoretical and simulation results described in the previous paper, an irradiation platform is built and irradiation experiments are conducted on the simulated mechatronics electronic system to observe whether the response of the simulated system to strong electromagnetic radiation interference is consistent with the previous analysis.

### 5.1. Test System

According to the experimental requirements, this test is configured with the test system shown in Figure 16. The main test equipment is as follows.

Transmitting antenna: since the test frequency range is 0.1 to 10 GHz, three antennas are selected: log-periodic antenna (0 to 2.4 GHz), double-ridge horn antenna (2.4 to 7.5 GHz), and horn antenna (7 to 18 GHz) to transmit electromagnetic waves of different frequencies.

Signal generator: Aglient Technologies E8257D, with a bandwidth of 10 MHz to 20 GHz, is used in test systems to generate excitation signal sources.

Two power amplifiers: bandwidths of 0 to 1 GHz and 1 to 18 GHz, respectively.

Field strength probe and measurement meter: frequency range 1 MHz~18 GHz, resolution 0.01 V/m, range 0.8~800 V/m.

During the experiment, the signal generator provides a continuous high frequency wave signal, which is fed into the transmitting antenna through two RF amplifiers of different frequency bands to provide the high-frequency and high-power electromagnetic radiation environment required for the test. The bracket where the simulated product is placed is made of plastic, which will not affect the distribution of the electromagnetic field. The transmitting antenna faces an open and unobstructed environment to form a reflection-free planar electromagnetic wave.

### 5.2. Experimental Subject

To meet the experimental requirements, a simulated mechatronics product was created, and the mechatronics product circuit only included the actuation circuit and the power supply. For the observability of the experiment, a buzzer was used to indicate whether the electromechanical circuit system is disturbed by electromagnetic radiation. If the simulated circuit is subjected to strong electromagnetic interference during the electromagnetic irradiation test, the buzzer beeps.

The experiment aims to verify the effectiveness of microwave absorbing material on electromagnetic interference suppression, so the simulated tests are divided into two categories, as shown in Figure 17; that is, the inner wall of the product housing is mounted with or without microwave absorbing material. During the test, the execution circuit is in working condition, and the upper face of the product shell is facing the direction of plane wave propagation. At the end of the irradiation, check whether the actuator circuit can continue to work properly.

### 5.3. Strong Electromagnetic Field Irradiation Test

During the experiment, a strong electromagnetic field irradiation test was conducted in the range of 0.1 to 10 GHz using a swept continuous wave. The test steps are as follows.
(1)Generate a strong electromagnetic field environment with the transmitting antenna, select the location of the product to be measured, and then use the field strength meter to measure whether the electric field strength of the location under different frequency points meets the requirements (according to 200 V/m, 500 V/m, 800 V/m, gradually increasing the field strength).(2)The product with the actuation circuit is placed in the optimal energy coupling direction, then start irradiation, with a dwell time of 2 s at each frequency point. Observe the buzzer to see if it beeps.(3)After each frequency point is irradiated, the actuation circuit is checked for continued proper operation.

No interference was observed at 200 V/m and 500 V/m for the simulated product without absorbing material, and the test results were shown in Table 4 when the irradiation field strength was 800 V/m. The simulated products with absorbing materials were not observed to interfere at all three irradiation field strengths.

The experimental results show that the simulated test product without absorbing material is interfered with by electromagnetic radiation at multiple frequency points, causing malfunctions or abnormal circuit functions, and returns to normal after a period of time. From the simulation results in Section 4, it is clear that the mis-operation or abnormal circuit function near the 5 GHz frequency point is mainly caused by the cavity resonance. The abnormal occurrence of the test circuit may be due to the rapid rise of the internal temperature of the semiconductor devices (such as microprocessor or NMOS device) under the condition of strong electromagnetic irradiation, which leads to transient failure or slight damage of the device, resulting in a similar short-circuit phenomenon in the circuit system with small internal resistance. It is known from previous studies [47] that permanent failure of semiconductor devices is due to the accumulation of heat inside the device, which leads to an increase in temperature to reach the melting point of the material. In the irradiation test of this paper, the continuous irradiation time of each frequency point is only 2 s, which means that the action time of the strong electromagnetic field on the circuit is also relatively short. The device may return to its normal performance after a brief failure for a period of time. Other frequency points do not indicate obvious interference after electromagnetic irradiation, which may be due to the limitation of the experimental conditions. During the experiment, the distance between the irradiation frequency points is large, and the ideal frequency sweeping irradiation is not realized. Therefore, it is very likely that the frequency points that may cause the failure of the product execution circuit are missed. The simulated test product with microwave absorbing material is not observed to interfere with the phenomenon, which verifies that the microwave absorbing material can effectively suppress the coupling effect of a strong electromagnetic field and has a good electromagnetic protection effect on the system circuit.

## 6. Conclusions

The MWCNTs/FCI/PU composite absorbers were prepared by mechanical blending, in which S1.5 (1.5 wt.% of MWCNTs) has good microwave absorption performance. For a thickness of 3 mm, the matching was optimum, with a minimum reflectivity of −29 dB at 5.2 GHz. The minimum reflectivity reaches −15.6 dB at a thickness of 1.5 mm, and the bandwidth with reflectivity less than −10 dB is 5.7 GHz. Importantly, the composite exhibits a good electromagnetic shielding effect when the carbon nanotube mass fraction and material thickness are only 1.5 wt.% and 1.5 mm, respectively, which are lower than previously studied MWCNTs-based composite absorbing material. Finally, the role of composite absorbing material electromagnetic protection was tested by both theoretical simulation and irradiation test, and the results both verify that absorbing material can significantly improve the anti-strength electromagnetic interference performance. Therefore, the MWCNTs/FCI/PU composite is a promising light-weight and efficient absorbing material.

## Figures and Tables

**Figure 1 materials-15-05690-f001:**
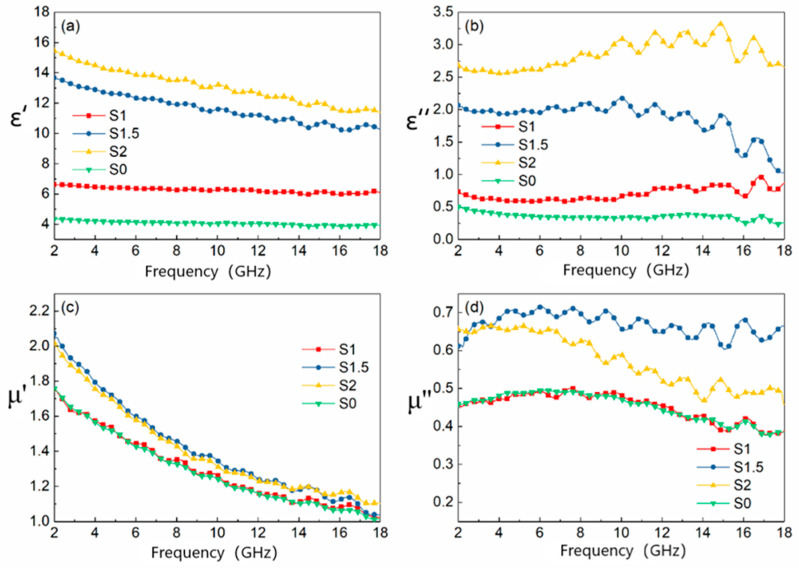
Electromagnetic parameters of four samples. (**a**) Real part of permittivity; (**b**) imaginary part of permittivity; (**c**) real part of permeability; (**d**) imaginary part of permeability.

**Figure 2 materials-15-05690-f002:**
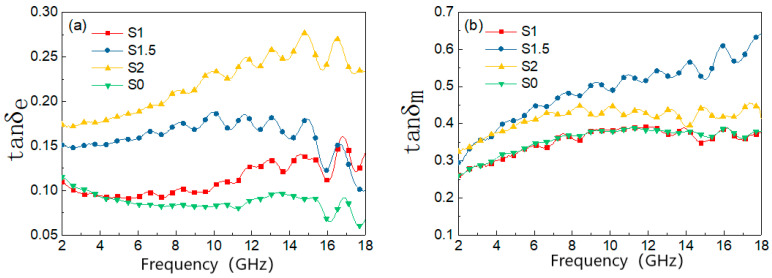
Dielectric and magnetic losses of composite materials. (**a**) Dielectric loss; (**b**) magnetic loss.

**Figure 3 materials-15-05690-f003:**
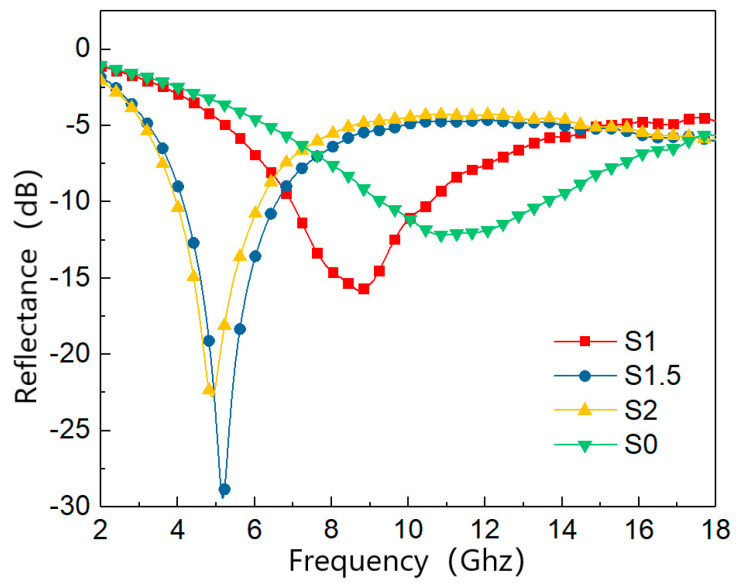
The RL of four samples with 3 mm thickness.

**Figure 4 materials-15-05690-f004:**
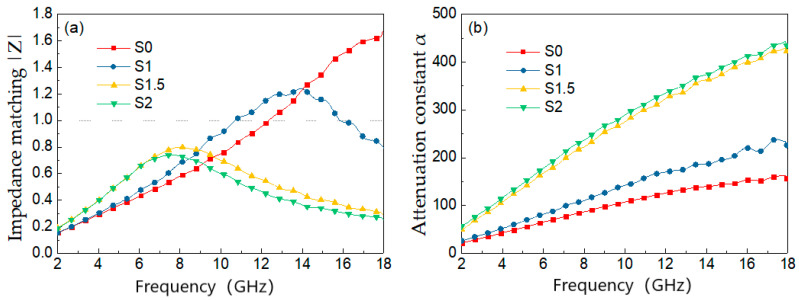
(**a**) Impedance matching ratio; (**b**) attenuation constant.

**Figure 5 materials-15-05690-f005:**
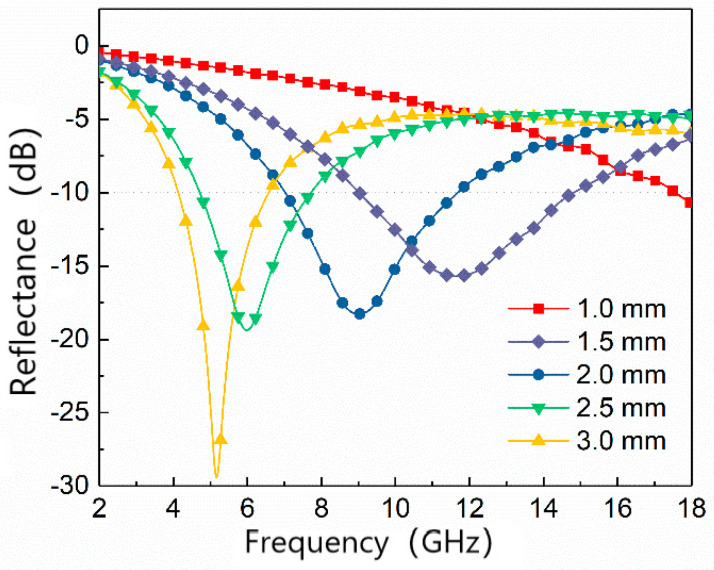
The RL of S1.5 at various thicknesses.

**Figure 6 materials-15-05690-f006:**
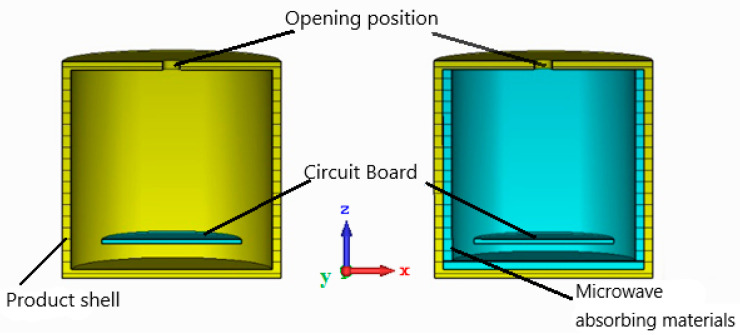
Approximate model of mechatronics product. Before mounting absorbing material (**left**); after mounting absorbing material (**right**).

**Figure 7 materials-15-05690-f007:**
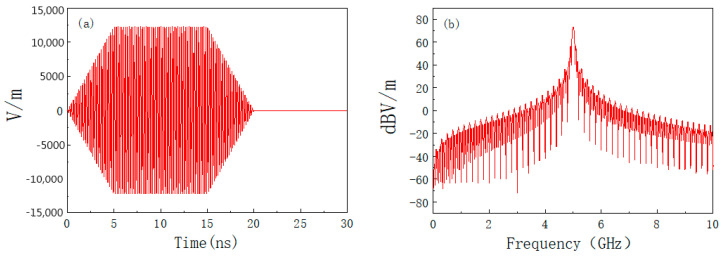
HPM waveform: (**a**) time domain; (**b**) frequency domain.

**Figure 8 materials-15-05690-f008:**
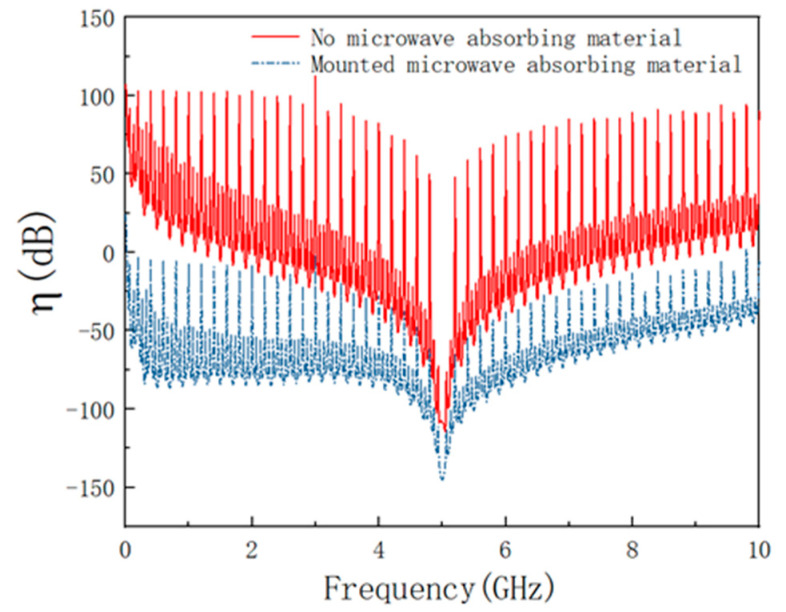
Coupling coefficient before and after mounting absorbing material under HPM.

**Figure 9 materials-15-05690-f009:**
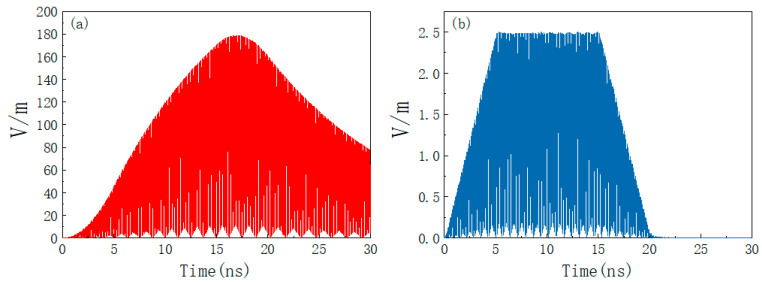
Electric field strength at the center of the product cavity under HPM amplitude illumination. (**a**) No absorbing material; (**b**) placement of absorbing material.

**Figure 10 materials-15-05690-f010:**
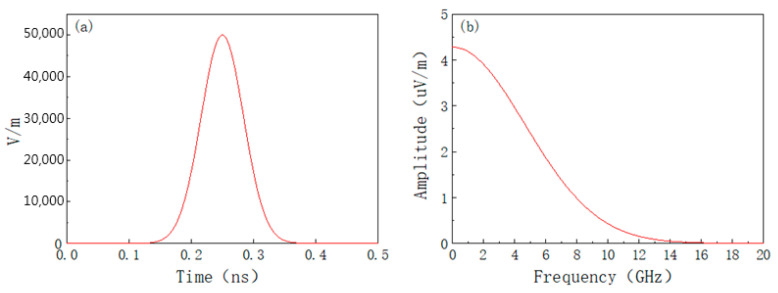
UWB waveform: (**a**) time domain; (**b**) frequency domain.

**Figure 11 materials-15-05690-f011:**
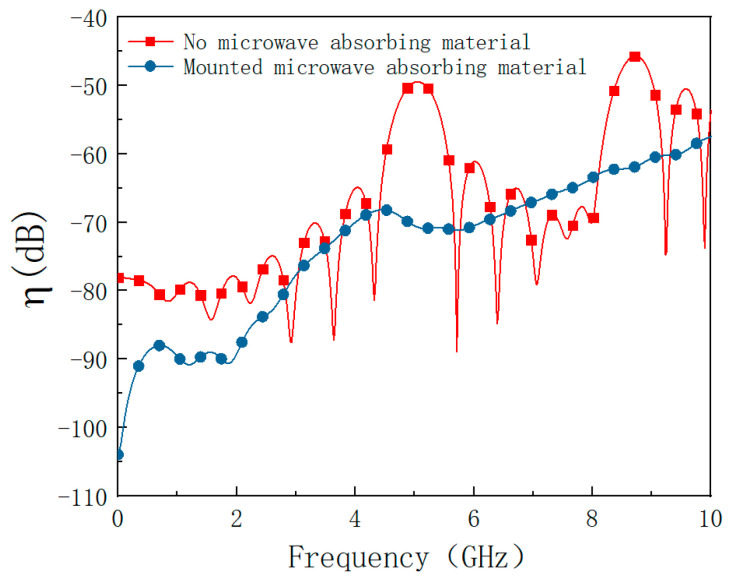
Coupling coefficient before and after mounting absorbing material under UWB.

**Figure 12 materials-15-05690-f012:**
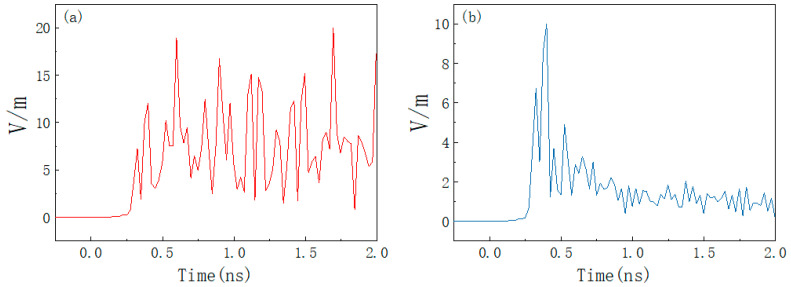
Electric field strength at the center of the product cavity under UWB amplitude illumination. (**a**) No absorbing material; (**b**) placement of absorbing material.

**Figure 13 materials-15-05690-f013:**
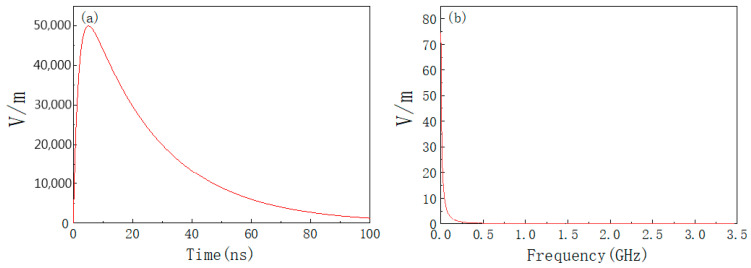
NEMP waveform. (**a**) Time domain; (**b**) frequency domain.

**Figure 14 materials-15-05690-f014:**
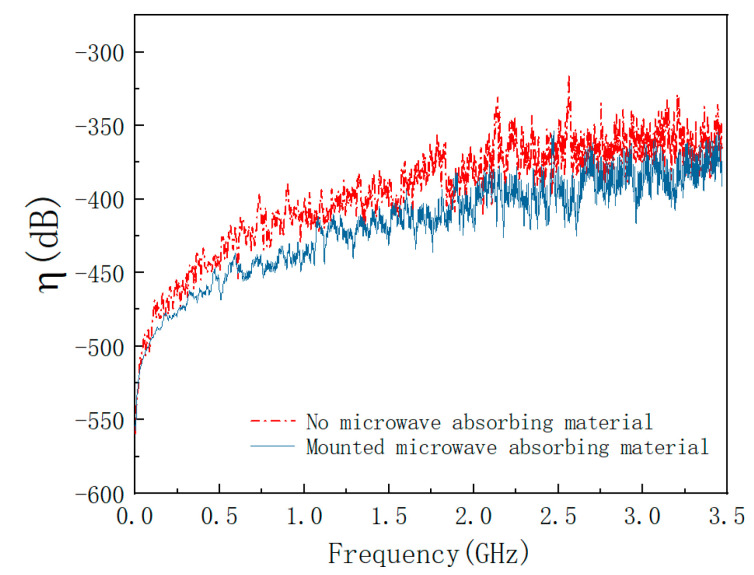
Coupling coefficient before and after mounting absorbing material under HEMP.

**Figure 15 materials-15-05690-f015:**
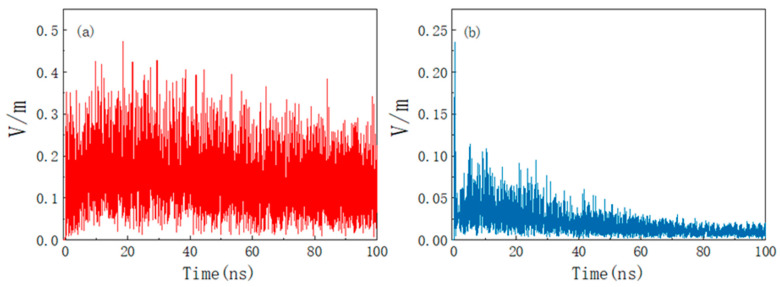
Electric field strength at the center of the product cavity under NEMP amplitude illumination. (**a**) No absorbing material; (**b**) placement of absorbing material.

**Figure 16 materials-15-05690-f016:**
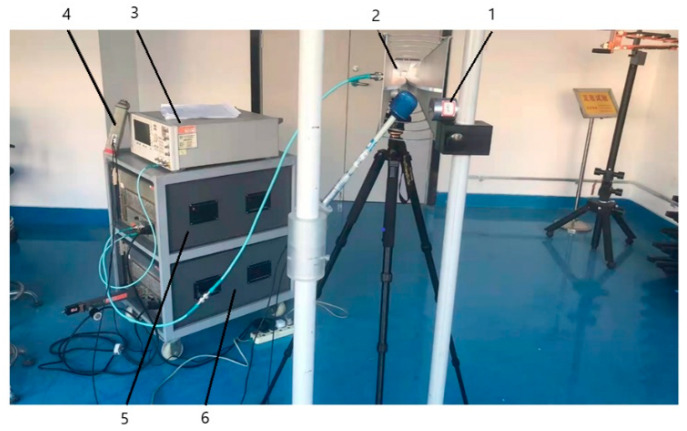
Strong electromagnetic field irradiation experimental device. 1. analog mechatronics product; 2. transmitting antenna; 3. signal transmitter; 4. field strength measurement meter; 5. 0~1 GHz power amplifier; 6. 1~18 GHz power amplifier.

**Figure 17 materials-15-05690-f017:**
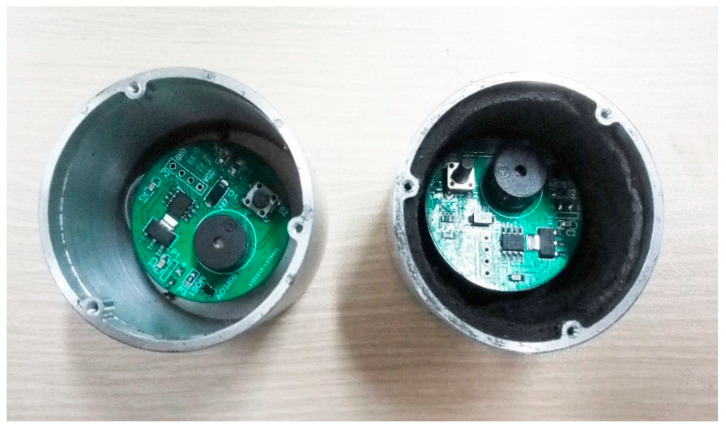
Analog mechatronics product. Without absorbing material (**left**). With absorbing material (**right**).

**Table 1 materials-15-05690-t001:** Summary of research status of MWCNTs-based composite absorbing materials.

Year of Publication	Absorber	Technique	Minimum RL (Frequency, Thickness)	Reference
2017	CoFe_2_O_4_/MWCNTs	co-precipitation method	−32.5 dB (8.88 GHz, 2 mm)	[14]
2021	Co/MWCNTs/polyethylene	situ polymerization of ethylene	−55 dB (5.2 GHz, 4 mm)	[17]
2020	CobaltSulphate/MWCNTs	three-step method	−30.22 dB (11.8 GHz, 3 mm)	[18]
2020	RGO/MWCNTs/CeO_2_	one-pot hydrothermal route	−59.3 dB (4.6 GHz, 4.5 mm)	[19]
2017	MWCNT/graphene foam	solvothermal method	−39.5 dB (11.6 GHz, 10.0 mm)	[20]

**Table 2 materials-15-05690-t002:** Preparation of the required chemical reagents and materials.

Reagent Name	Manufacturer
multi-walled carbon nanotubes	Suiheng Technology Co., Ltd., Shenzhen, China
flake carbonyl iron particles	Blue Magnetic New Material Technology Co. Ltd., Guangzhou, China
polyether polyol	Hagibis Co., Ltd., Beijing, China
polyisocyanate	Hagibis Co., Ltd., Beijing, China
resin dispersant	XFNANO Materials Tech Co., Ltd., Nanjing, China

**Table 3 materials-15-05690-t003:** Material proportions.

Sample No.	MWCNTs (wt.%)	FCI (wt.%)	Polyol (wt.%)	Isocyanate (wt.%)	Dispersant (wt.%)
S0	0	70	21.4	8.6	0
S1	1	70	17.8	7.2	4
S1.5	1.5	70	16	6.5	6
S2	2	70	14.3	5.7	8

**Table 4 materials-15-05690-t004:** The results of the irradiation test for each frequency point of the simulated product without absorbing material (800 V/m).

Frequency (GHz)	Test Result	Frequency (GHz)	Test Result	Frequency (GHz)	Test Result
0.1	-	3.7	-	6.0	-
0.3	-	3.9	-	6.1	-
0.5	-	4.1	-	6.2	-
0.7	-	4.3	Beep	6.3	-
0.9	-	4.5	-	6.4	-
1.1	-	4.6	-	6.7	-
1.3	-	4.7	-	7.0	-
1.5	-	4.8	Beep	7.3	-
1.7	-	4.9	*	7.6	-
1.9	-	5.0	*	7.9	-
2.1	-	5.1	Beep	8.2	-
2.3	-	5.2	-	8.5	*
2.5	-	5.3	-	8.8	-
2.7	-	5.4	-	9.1	-
2.9	-	5.5	-	9.4	-
3.1	-	5.6	-	9.7	-
3.3	-	5.7	-	10	-
3.5	-	5.8	-		

* Indicates that the buzzer keeps sounding after the circuit is re-powered and resumes after a period of time.

## Data Availability

The data used to support the findings of this study are available from the corresponding author upon request.

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
