# Peer review of "Microwave Absorption Properties of Multi-Walled Carbon Nanotubes/Carbonyl Iron Particles/Polyurethane Foams"

_materials, 2022, doi:10.3390/ma15165690_

Round 1

Reviewer 1 Report

Comments to the author

In this paper, the authors report composite foam type absorbing materials with different multi-walled carbon nanotube (MWCNTs) using polyurethane foam as the substrate and MWCNTs and flaked carbonyl iron powder as absorbers to improve the microwave absorption performance of absorbing materials. The electromagnetic properties of the materials have been characterized and analyzed with simulation and experiment. The paper is generally nicely written and the results are promising. I would like to recommend the paper for publication provided addressing the following issues:

1.      In literature review, few of the studies are comparatively outdated. I recommend to add more recent studies related to the absorbing materials and multi-walled carbon nanotube (MWCNTs). There should be at least 10-15 studies not older than 2017.

2.      Please add a Table in section summarizing the literature review by considering the year of publication, the type of absorbing materials and MWCNTs used, technique, major findings and finally compare with your work. I am sure this comparison/ table will definitely elevate the interest of reader’s community in this work. Thanks

3.      The title of section 2 should be revised. Just “experimental” feels awkward. You can use “Material Preparation”, “Sample Preparation” or any other suitable title.

4.      The label of x-axis in Fig.1-5, 8-15 should be like “Frequency (GHz)” or Time (ns) etc. Please revised all the figures. Also, the font of label of y-axis in these figures is difficult to read. Please increase the font size.

5.      There are various typos, spelling issues, syntax errors etc. at various places in the paper. For, examples,

·         Abstract Line 12: “Then take a mechatronics product…..” This sentence does not make any sense. Please revised this sentence. Similarly, Line 13-14, “…….strong EM irradiation simulation and experiments”. What the authors want to convey? Not getting the meaning. Please revise the abstract and briefly state the findings.

·         Section 1 Line 30-36, the sentences are full with “and”. Please revise these lines. Similarly, Line 90-92, don’t use “the” prefix with all methods/ techniques.

·         Section 2 Line 121, 122, revise these sentences as there are grammatical errors.

·         Section 3 Line 166, revise this paragraph because this is not abstract where you need to write “In this paper”.

These are only few examples, but the paper is full of plenty of errors. I strongly recommend to thoroughly proofread the complete text. You may take help from English language expert of online formatting tool. 

Author Response

Dear reviewer, we have revised the article based on your comments. Please see the attachment for details.

Author Response

(The authors gave the same response as above.)

Reviewer 3 Report

The authors presented a flexible porous composite absorbing material with MWCNTs, FCI particles as absorbers, and PU foam as the matrix. The composites' complex permittivity and complex permeability were measured by the T/R method and the materials' electromagnetic and microwave absorption properties were investigated. The results are interesting and the methods are sufficient. By the way, the introduction needs to discuss in more detail. Some comments are as below:

-why flaked carbonyl iron powder was used as an absorber? What are the features of this compound?

-please add carbon nanotube to the keyword

-it suggested to add the following refs to enrich your introduction: https://doi.org/10.1002/admt.202100698, https://doi.org/10.1515/ntrev-2021-0087, https://doi.org/10.1088/1361-6528/ac05e7

-what about the morphology of prepared Structures? At least an electron microscopy analysis must be added

-conclusion didn’t provide the goal and conclusion of this study. Please rewrite

-it is better to add a graphical abstract to summarize the goal of this study

Author Response

(The authors gave the same response as above.)

Round 2

Reviewer 2 Report

Dear Authors, thank you for your updated version of the manuscript. I am happy with the changes made.

Author Response

Dear reviewer, we are glad to receive your reply and thank you for your approval of our revised content. We wish good health to you, your family, and community. Your careful review has helped to make our study clearer and more comprehensive.

Reviewer 3 Report

The authors didn't address all my comments. therefore another round of revision is required

for example I suggested the authors discuss some new papers in the field but they didn't add the following refs

-it is suggested to add the following refs to enrich your introduction: https://doi.org/10.1002/admt.202100698, https://doi.org/10.1515/ntrev-2021-0087, https://doi.org/10.1088/1361-6528/ac05e7

Author Response

Thanks for the reviewer’s careful review. First of all, we apologize to the reviewer because we did not quote the references you suggested to add in the reply and did not explain them in the article. For the reviewer's suggestion, we study and discuss each item in detail. We also study and summarize the three references you suggested to add.

The first article titled "Acoustic Metamaterials for Noise Reduction: A Review". This article provides a comprehensive overview of the development of acoustic metamaterials, summarizing the basic classification, underlying physical mechanism, application scenarios, and emerging research trends for both passive and active noise-reduction metamaterials. The second is titled "Low-voltage and fast-response SnO2 nanotubes/perovskite heterostructure photodetector" article, which mainly introduces SnO2 NTs/perovskite heterostructure materials for high-performance photodetectors, making the detectors exhibit fast response/recovery speed and a wide optical response range. The third article, entitled "On the rheological properties of multi-walled carbon nano-polyvinylpyrrolidone/silicon-based shear thickening fluid", examines the rheological properties of shear thickening fluid (STF) enhanced by additives such as multi-walled carbon nanotubes (MWCNTs), poly-vinylpyrrolidone (PVP), and nano-silica (SiO2) at different mass fraction ratios. These three articles are related to carbon nanotube composites, but their application fields are different. The research content of this paper is about composite absorbing materials of multi-walled carbon nanotubes. The research direction in the literature is quite different from this paper, so the above literature is not cited. But we will study these papers carefully and refer to them in subsequent related research. In response to the reviewers' mention of insufficient content in the manuscript review, we have sorted out the articles on multi-wall carbon nanotube absorbing materials in recent years, deleted some relatively old literature in the manuscript, and added the latest research status. Finally, thanks again to the reviewer for your comments. Your careful review has helped to make our study clearer and more comprehensive.